# Use of Raman and Raman optical activity to extract atomistic details of saccharides in aqueous solution

**Vladimír Palivec**[1], **Christian Johannessen**[2], **Jakub Kaminský**[1], **Hector Martinez-Seara**[1] *

**1** Institute of Organic Chemistry and Biochemistry, Czech Academy of Sciences, Prague, Czech Republic,
**2** Department of Chemistry, University of Antwerp, Antwerp, Belgium

* hseara@gmail.com

**Data Availability Statement:** All relevant data are within the paper, its Supporting information files, and on Zenodo at DOI: 10.5281/zenodo.5791738.

## Abstract

Sugars are crucial components in biosystems and industrial applications. In aqueous environments, the natural state of short saccharides or charged glycosaminoglycans is floating and wiggling in solution. Therefore, tools to characterize their structure in a native aqueous environment are crucial but not always available. Here, we show that a combination of Raman/ROA and, on occasions, NMR experiments with Molecular Dynamics (MD) and Quantum Mechanics (QM) is a viable method to gain insights into structural features of sugars in solutions. Combining these methods provides information about accessible ring puckering conformers and their proportions. It also provides information about the conformation of the linkage between the sugar monomers, i.e., glycosidic bonds, allowing for identifying significantly accessible conformers and their relative abundance. For mixtures of sugar moieties, this method enables the deconvolution of the Raman/ROA spectra to find the actual amounts of its molecular constituents, serving as an effective analytical technique. For example, it allows calculating anomeric ratios for reducing sugars and analyzing more complex sugar mixtures to elucidate their real content. Altogether, we show that combining Raman/ROA spectroscopies with simulations is a versatile method applicable to saccharides. It allows for accessing many features with precision comparable to other methods routinely used for this task, making it a viable alternative. Furthermore, we prove that the proposed technique can scale up by studying the complicated raffinose trisaccharide, and therefore, we expect its wide adoption to characterize sugar structural features in solution.

## Author summary

Saccharides are an essential part of many biosystems. Not only their identification but also their structural characterization is crucial to understanding their role. For example, animal cells possess a surrounding aqueous layer rich in sugars. Although its sugar content is known, the structures formed by these sugars in such a flexible and mobile environment are lesser understood. A trustworthy characterization of their atomistic structure will foster our knowledge and contribute to drug development. This characterization is also

**Funding:** HMS, JK, and VP acknowledge support from the Czech Science Foundation (19-19561S). The funders had no role in study design, data collection and analysis, decision to publish, or preparation of the manuscript.

**Competing interests:** The authors have declared that no competing interests exist.

crucial in developing accurate computer models needed to study such environments. Unfortunately, many available structure characterization techniques are inadequate when applied to flexible molecules such as saccharides. Therefore, new methods are welcomed. We use molecular dynamics simulations and quantum chemical calculations to extract structural data of sugars in solution when compared against experimental Raman/ROA/NMR spectra. This combination of techniques allows interpreting aforementioned spectra to extract structural features of saccharides. We get, for example, the molecular structure of sugar rings (puckering confirmation) or the preferred orientation of bonds connecting two sugar units (glycosidic bonds). We can determine the presence of different sugar moieties in solutions and their proportions. Overall, this technique should contribute to the understanding of sugars in their native aqueous environment.

This is a *PLOS Computational Biology* Methods paper.

## Introduction

Carbohydrates are a key and diverse molecular group present in all living matter. It is crucial in biological functions and structure, including plant scaffold (e.g., cellulose), primary energy storage reservoir (e.g., starch/glycogen), or genetic code backbone (e.g., deoxyribose). The function and structural properties of sugars in these examples are well understood. Other roles of sugars have just started gathering attention, such as the role of branched polysaccharides attached to proteins [1] or glycosaminoglycans found in glycocalyx [2]. Glycosaminoglycans with their charged groups and hydroxyl groups are exceptionally hydrophilic and flexible, exploring a vast number of conformational states [3]. Proper understanding of glycosaminoglycans' structural properties in their functional environment, i.e., aqueous solution, is crucial to understanding their function [4]. However, correct and detailed structural determination of saccharides has proved to be challenging. Although there are many well-established techniques to extract atomistic details, most were developed for proteins or nucleic acids, and cannot be easily applied to flexible molecules like saccharides. For example, saccharides are very hard to crystallize, or often the obtained crystals are of insufficient quality for X-ray crystallographic experiments [5]. Moreover, even successful determination of the 3D structure from X-ray often yields incomplete structure due to the mobility of the saccharide moiety [6], or the obtained structural details differ from the solution state [7, 8]. Saccharides also lack good natural chromophores hindering their study using the UV-Vis fluorescence spectroscopies. Nowadays, the methods of Nucleic magnetic resonance (NMR) are mainly used to study saccharides in solution; however, even NMR has its problems such as overlapping signals or the need for isotopically labeled compounds. All in all, saccharides are challenging to study using traditional structural techniques, which leads to an effort in exploring new structural methods for their characterization. One promising method is Raman and Raman Optical Activity (ROA) [9] spectroscopy. ROA is a Raman-based spectroscopic method where the difference in Raman spectra when using right($I_R$)/left($I_L$) circularly polarised light spectra is recorded, i.e., $I(ROA) = I_R - I_L$. The advantage of Raman/ROA techniques when studying saccharides is a transparent vibrational spectral region between 75 and 3100 $cm^{-1}$, as compared to more traditional infrared spectroscopy (IR) coupled with vibrational circular dichroism (VCD), which

offer limited information due to the strong water absorption. Moreover, Raman/ROA spectroscopies are very sensitive to conformational changes in saccharides both in the fingerprint [10] (75–1600 cm$^{-1}$) and CH stretching region [11] (2700–3100 cm$^{-1}$). Although Raman/ROA spectroscopies are readily accessible experimentally, their experimental spectra alone provide limited structural information due to their complexity and the substantial effects of the surrounding water molecules. Simulations can disentangle the crowded spectra and extract the underlying structural information. For example, the approach was used to investigate differences of two epimers, glucose and galactose [12], and to study their vibrational properties. It also allows studying mixtures such as glucose/mannose [13], which spectra can be decoupled to individual moieties. Another example is identifying conformers in flexible sugars such as gluconic acid that contribute to the experimental spectra [14]. It has also been applied to study sorbose/glucose in the aqueous solution in an extended vibrational range (75–3100 cm$^{-1}$) [11], where the effects of small conformational changes of $CH_2OH$ group were linked to the overall shape of spectra. The method is not limited only to monosaccharides as it has been applied to the study of agarose trisaccharides in the gel phase to gain insight into its packing [15] or as in the study of differently linked mannobioses [16]. The ROA spectra were also used to characterize glucose upon its phosphorylation and study the influence on conformational preferences [3]. All of these studies show that Raman/ROA spectroscopies benefit from coupling to computer simulations and that there is and interest of utilizing the techniques to study saccharides.

The accuracy of simulation protocols limits the extraction of structural features, and therefore, significant effort has been devoted to developing reliable and economical computational protocols. Although there exist other calculation protocols to simulate Raman/ROA spectra [17], most of the state-of-the-art protocols nowadays consist of ensemble averaging many structures, usually tens to hundreds [12, 16, 18, 19], generated with MD simulations. The microscopic structure of the first solvation layer (3–3.6 Å cutoff) was found to be very important [10–12, 14, 18–22], while the long-range polarization can be modeled either with continuum solvation methods or using a larger shell of explicit water molecules (10–12 Å cutoff) [13, 19]. The spectra are then calculated for solute-solvent clusters at the DFT level, usually using the B3LYP or BPW91 functionals, with a sufficiently large basis set [10, 23], e.g., 6–311++G$^{**}$ or aug-cc-pVTZ. Since Raman/ROA tensor invariant calculations demand diffuse basis set functions, while force field calculations generally do not, performing split calculations have been proposed to speed up the calculations. For example, the rDPS basis set [24] consists of calculating the force field part with 6–31G$^{*}$ basis set, while the tensor invariants with 3–21++G basis set augmented with a semi-diffuse p functions with an exponent of 0.2 on hydrogen atoms. For the final spectra calculations, the ensemble generated with MD needs to be optimized at the quantum mechanics (QM) level of theory; however, complete optimization is detrimental [10]. Partial optimization approaches such as the Normal modes optimization [25] or optimizing only the central molecule while freezing the rest of the environment [12] are used. It was recently shown that a simple ten-step optimization without any restrains is sufficient to capture the sought effects while being very simple [10]. Usually, some empirical correction of calculated vibrational frequencies is performed [16], either as a simple one-factor scaling or using more elaborate methods [10]. Many of the presented important factors were recently thoroughly investigated on a set of six model monosaccharides, and as a result, an accurate and cost-efficient computational protocol was proposed [10]. There, the interest was to study various technical aspects of simulation of Raman/ROA spectra, while this work focuses purely on the application of this protocol to interpret the experimental data and access structural details of saccharides in an aqueous solution.

In this paper, we explore the boundaries of using Raman/ROA spectroscopies coupled with computer simulations to extract structural details of saccharides in an aqueous solution. We

show that rotation around glycosidic bonds of disaccharides leads to profound spectral changes that unambiguously identify the most abundant rotamers. Moreover, we study populations of ring conformers using methyl-β-D-glucuronic acid as a model system aiming to distinguish between various puckered conformations and their relative weights [26]. The current applicability of the simulation protocols to larger saccharides is tested on a raffinose trisaccharide. We also extract anomeric equilibrium values on a set of well-characterized monosaccharides and this approach is then extended to binary mixtures of known compositions to determine the experimental mixing ratios. Lastly, we explore the effect of intermolecular interactions on a model system of two stacked monosaccharides. When possible, we compare our results to MD/NMR data to highlight that Raman/ROA brings additional structural information and that it can be used as a complementary technique to study saccharides in a water environment.

## Methods

### Target saccharides

We have chosen six pyranose monosaccharides: D-glucose (Glc), D-glucuronic acid (GlcA), 2-acetamido-2-deoxy-D-glucose (aka *N*-acetyl-D-glucosamine; GlcNAc), and their methyl glycosides, methyl-β-D-glucoside (MeGlc), methyl-β-D-glucuronide (MeGlcA), and methyl-2-acetamido-2-deoxy-β-D-glucoside (MeGlcNAc) to study how Raman and ROA spectroscopies can be used to extract structural details of saccharides in solution (Fig 1). We have also included four glucose or mannose containing disaccharides: α-D-glucopyranosyl-(1→1)-α-D-glucopyranose (trehalose), methyl-1α-2α-mannobiose (M12), methyl-1α-3α-mannobiose (M13), and methyl-1α-6α-mannobiose (M16) to study the effects of rotation around glycosidic bonds on Raman/ROA spectra (Fig 2; explicit atom definition of dihedral angles in S1 File). An important feature of these disaccharides is that they are non-reducing, and therefore, they do not epimerize to an α/β anomeric mixture. Lastly, we have also included a trisaccharide—raffinose (Gal(α1–6)Glc(α1–2β)Fruf) (Fig 2, bottom), to test the performance of the simulation protocol on a larger moiety.

### Simulation details

Simulation of Raman/ROA spectra and connection to experimental data consist of several steps. 1) A representative ensemble of structures is generated using molecular dynamics simulations. 2) Raman/ROA spectra are calculated for each of the structures using quantum chemical methods, obtaining ensemble-averaged spectra. 3) We score the similarity of simulated spectra with experiment, and when applicable, we find the best fit of simulated data to the Raman/ROA spectra. 4) When possible, we compare our calculations to NMR data. Underneath, we explain each of these steps.

   **Ensemble generation.**   Molecular dynamics (MD) simulations were used to generate representative ensembles used to simulate spectra of studied saccharides. All MD simulations were performed using the Gromacs-2018.4 [29] simulation package patched with Plumed 2.5 [30]. The systems of interest were simulated at ambient conditions (p = 1 bar and T = 300 K) using Glycam-6h [31] (sugars) and OPC3 [32] (water) models. The rest of the simulation parameters are rather standard and are thoroughly described in S1 File. We performed three types of calculations. 'Unbiased MD' simulations from which representative ensemble of structures were obtained, for which subsequently Raman/ROA spectra was calculated. 'Free energy' simulations which used well-tempered metadynamics MD [33] to calculate the free energy profile along studied coordinate such as glycosidic dihedral angles or puckering coordinates. We use those to find local minima on the free energy profiles for which we performed a third

**Fig 1. Investigated monosaccharides.** The Figures were prepared with ChemSketch [27] and edited with Inkscape [28].

type of a simulation. These are 'biased MD' simulations that restrained the system around selected local minima using harmonic position restrains for which ensemble average spectra were calculated. The ensemble averaged spectra for each minima were used to best fit experimental data therefore estimating their contributing weights to the overall spectra. Representative ensembles were obtained by uniformly extracting 250 structures from the 'biased MD' and 'unbiased MD' simulations. When studying Binary Mixtures and Crowding Effects 500 structures were used. Table 1 summarizes all performed simulations in this work. For further details see Methods in S1 File.

**Calculation of Raman/ROA Spectra.**   Raman/ROA spectra of simulated ensembles were calculated using a previously developed `hybrid` simulation approach [10]. In brief, this simulation protocol is a QM/MM based method using the Gaussian16 program package [34] that consists of extracting snapshots from the MD trajectory, where the central molecule together with surrounding water molecules (3 Å cutoff) are kept. The water molecules are treated at the MM level of theory, while the solute is calculated at the B3LYP/6–311++G$^{**}$ level of theory using the ONIOM method with electrostatic embedding [35, 36]. TIP3P [37] and Glycam-6h [31] force field parameters were used to describe the MM part of the ONIOM method. The long-range solvation effects are taken care of by using the COSMO implicit solvation model. After preparation, the system is optimized in ten steps unrestrained optimization. After optimization, we use the far-from-resonance coupled-perturbed Kohn-Sham theory [38, 39] at

**Fig 2. Investigated oligosaccharides.** Investigated disaccharides: trehalose, methyl-1α-2α-mannobiose (M12), methyl-1α-3α-mannobiose (M13), and methyl-1α-6α-mannobiose (M16), together with defined glycosidic dihedral angles $\phi_1$, $\phi_2$, and $\phi_3$ (explicit atom definitions in S1 File). Bottom raffinose trisaccharide.

532 nm (SCP180) and harmonic approximation to calculate vibrational frequencies and Raman/ROA intensities. All calculated vibrational frequencies are rescaled using a frequency scaling function $\phi(\tilde{v}, a, b, c, d) = \phi(\tilde{v}, 0.982, 1.00, 15, 1210)$ [10]. Calculated intensities were corrected for the temperature factor at 300 K and convoluted using Lorentzian with a full width at half maximum (FWHM) $\Gamma = 7.5$ cm$^{-1}$.

To asses the quality of simulated spectra we used the overlap integral $S$ defined in Eq (1).

$$S = \frac{\int_{\tilde{v}_{min}}^{\tilde{v}_{max}} I_{sim}(\tilde{v}) I_{exp}(\tilde{v}) d\tilde{v}}{\sqrt{\int_{\tilde{v}_{min}}^{\tilde{v}_{max}} I_{sim}^2(\tilde{v}) d\tilde{v} \int_{\tilde{v}_{min}}^{\tilde{v}_{max}} I_{exp}^2(\tilde{v}) d\tilde{v}}}, \tag{1}$$

where $I_{sim}$ and $I_{exp}$ represent the simulated and experimental spectra that are to be compared. Notably, the calculation protocol compares Raman/ROA simulation spectra to experimental ones individually, i.e., it scales calculated intensities to match experimental using two scaling factors $\rho_{Raman}, \rho_{ROA}$. As an effect, the circular intensity difference (CID, CID $= \frac{I_{ROA}}{I_{Raman}}$) ratios are not necessarily preserved, but effectively the difference we observe is small ($\sim 2.5\%$). Note that this does not affect any structural predictions because these are evaluated using the overlap integral, which is agnostic to any scaling of intensities. Furthermore, in our method CID is easily accessible as it is just scaled by a known constant and therefore could potentially be used as an additional source of structural data [40]. For more details, refer to former work [10] and SI.

**Table 1. Performed MD simulations.** Summary of MD simulations performed in this work.

| Conformation of Glycosidic Bonds (trehalose, M12, M13, and M16) | |
|---|---|
| 1 unbiased MD | 500 ns |
| 2a free energy | metadynamics MD in $\phi_1/\phi_2/\phi_3$ glycosidic dihedral angles |
| 2b biased MD | $md1/md2/md3/md4/md5/md6$ conformer regions 200 ns |
| **Probing Puckering Conformations (MeGlcA)** | |
| 1 unbiased MD | 500 ns |
| 2a free energy | metadynamics MD in $q_x$, $q_y$, and $q_z$ puckering coordinates |
| 2b biased MD | $^1C_4/^4C_1/^OS_2/^1S_3$ puckering conformer regions 50 ns |
| **$\alpha/\beta$ anomeric equilibrium constant (Glc, GlcA, and GlcNAc)** | |
| unbiased MD | 500 ns for each of the $\alpha/\beta$ anomer |
| **Raffinose Trisaccharide** | |
| unbiased MD | 500 ns |
| **Binary Mixtures** | |
| unbiased MD | 500 ns for pure MeGlc and MeGlcNAc |
| **Crowding Effects** | |
| unbiased MD | 500 ns for a single MeGlc |
| unbiased MD | 500 ns for two MeGlc in close proximity[a] |

[a] using a flat bottom potential with a COM-COM distance $< 0.5$ nm

**The Best Fit to Experimental Raman/ROA Spectra.** The experimental spectra are calculated as a cumulative average of all spectra produced by any accessible state of the system during the experiment. We strive to find relevance of each of selected states and their contribution to the final spectra. For this, computationally, we first calculate ensemble average spectra for each state ($I_{i,Raman/ROA}^{sim}$) such as $\alpha/\beta$ anomer forms or various molecular conformations. We seek to find their contribution to the overall simulated spectra,

$$I_{Raman/ROA}^{sim}(\vec{A}) = \sum_{i=0}^{spectra} A_i \cdot I_{i,Raman/ROA}^{sim}, \tag{2}$$

where $A_i$ is the weight given to a specific state $i$ and $\vec{A}$ is the vector of weights. The set of weights $A_i$ is calculated based on the similarity of simulated ($I_{Raman/ROA}^{sim}$) and experimental spectra, i.e., by minimizing the following cost function $F$:

$$F(\vec{A}) = (1 - S_{Raman}(\vec{A}))^2 + (1 - S_{ROA}(\vec{A}))^2, \tag{3}$$

where $S_{Raman}(\vec{A})$ and $S_{ROA}(\vec{A})$ are overlap integrals defined in Eq 1. The robustness of found best fit solution, i.e., the uncertainty of obtained set of weights $\vec{A}$, is determined by numerically finding weights that fulfill $S_{Raman} + S_{ROA} > 0.99(S_{Raman}^{bestfit} + S_{ROA}^{bestfit})$. The spread of the weights values $\{A_i^{min}, A_i^{max}\}$ is reported as the uncertainty of the best fit solution.

**Calculating conformer weights using NMR $^3J$ spin-spin coupling constants.** The ensemble average $^3J$ coupling constants were determined using Karplus equation [41] for glycosidic bonds of M12 and M13 disaccharides using raw MD data. We also used QM calculations for glycosidic bonds of M12 and M13 disaccharides and puckering of MeGlcA. The QM derived coupling constants were calculated using optimized snapshots used for Raman/ROA calculations stripped of all water molecules at mPW1PW91/pc-J2 [42] level of theory, together with the CPCM continuum solvation. To speed up the calculations, only the Fermi contact

term was considered. As in Raman/ROA spectra calculations, we determined weights of each conformer region by minimizing $G$,

$$G(\vec{A}) = \sum_{i=0}^{coupling\ constants} ({}^3J_i^{sim} - {}^3J_i^{exp})^2, \tag{4}$$

where ${}^3J_i^{sim}$ is the ensemble averaged simulation spin-spin coupling constant,

$${}^3J_i^{sim} = \sum_{j=0}^{conformers} A_j \cdot {}^3J_j^{sim}. \tag{5}$$

We evaluated the robustness of the obtained best fit weights by considering 0.2 Hz deviation (reported experimental error [41, 43]) from the best fit solution, see SI. The spread of the weights values $\{A_i^{min}, A_i^{max}\}$ is reported as the uncertainty of the best fit obtained weights.

## Experimental section

D-Raffinose pentahydrate (raffinose) was purchased from Sigma. MeGlc, MeGlcNAc, and trehalose were purchased from Carbosynth. Backscattered Raman and Scattered Circular Polarization (SCP) ROA spectra of all studied saccharide samples were recorded using the ChiralRAMAN-2X (Biotools Inc.) spectrometer. Stock aqueous solutions of 1 M concentration were prepared by dissolution of raffinose, MeGlc, and MeGlcNAc in milliQ water. Only 0.42 M solution of trehalose was used for further measurements due to limited amount of the sample (10 mg / 70 $\mu$L). Three different mixtures (1:1, 1:3, and 3:1) of mixtures of MeGlc and MeGlcNAc were prepared from their 1 M stock solutions. The impurities causing undesirable fluorescence background were partially removed from the samples by activated carbon filtration. Residual fluorescence was quenched by leaving the sample in the laser beam for at least an hour prior to the experiment. Then the Raman and ROA spectra of 1M MeGlc, 1M raffinose, 0.42 M trehalose, and three mixtures of MeGlc/MeGlcNAc were recorded. We used the 532 nm green laser with the power at the head from 700 mW (trehalose) to 900 mW (raffinose, MeGlc/MeGlcNAc mixtures). The illumination period was always set to 2.5 s. Rectangular fused silica cell with approximately 3 mm optical path and 50 $\mu$L sample volume was used for all experiments. The Raman and ROA spectra were recorded simultaneously using average acquisition time of approx. 24 h for each sample. Water signal was subtracted from the Raman spectra. When necessary a polynomial baseline correction was performed to fix any residual background or baseline drifting. The wavenumber scale was calibrated using a neon lamp. Fluorescence standard material (SRM 2242; National Institute of Standards and Technology, USA) was used for the ROA and Raman intensity calibration.

The rest of the Raman/ROA experimental data was obtained from the literature, i.e., Glc, GlcA, and GlcNAc, as well as the spectra of pure MeGlcA and MeGlcNAc [10] and M12, M13, and M16 [16].

## Results

### Conformation along the glycosidic bonds

First, we investigate selected disaccharides to monitor changes in the Raman and ROA spectra upon the rotations around their glycosidic bonds connecting the two monosaccharide units. This linker usually possesses two degrees of freedom, the $\{\phi_1, \phi_2\}$ angles. Our workflow will be thoroughly explained using methyl-1$\alpha$-3$\alpha$-mannobiose as a example. The results using the same protocol for trehalose and methyl-1$\alpha$-2$\alpha$-mannobiose will be relegated to S1 File as they

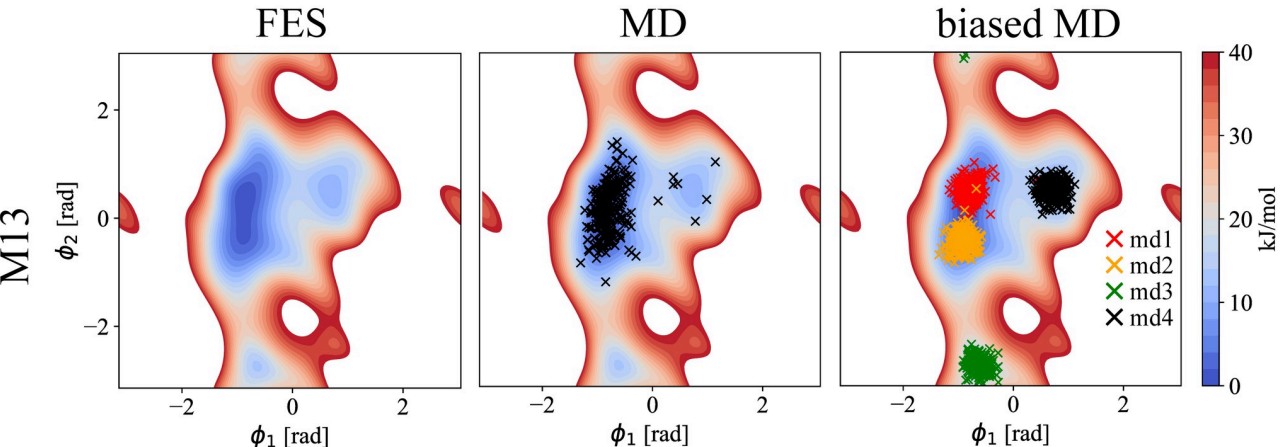

**Fig 3. M13 free energy surface and sampled structures.** Left: Calculated free energy surface (FES) of M13 disaccharide in $\{\phi_1, \phi_2\}$ dihedral angles. Middle: Calculated FES, together with 250 extracted structures from unbiased 500 ns MD simulations (MD). Right: Calculated FES, together with 250 extracted structures for each biased 200 ns MD simulation (biased MD; *md*1/*md*2/*md*3/*md*4; restrain $\{\phi_1, \phi_2\}$ values in Table 2). White regions represent area with the free energy >40 kJ/mol.

are similar. We conclude this section by discussing methyl-1α-6α-mannobiose. The 1→6 linked disaccharide is a rather special case as it possesses an additional degree of freedom. Note that conformational preferences of differently linked mannobioses have been studied previously under various conditions and using different methods for saccharides in their free states or as bound to lectins [44–48]. Similarly, also trehalose have been thoroughly studied predominantly using NMR and/or MD simulations [49–51].

**Conformation of Methyl-1α-3α-mannobiose (M13).** The left panel of Fig 3 shows the calculated free energy surface profile (FES) around $\{\phi_1, \phi_2\}$ dihedral angles for M13. The middle panel overlays the FES with the $\{\phi_1, \phi_2\}$ values obtained for 250 structures extracted from unbiased 500 ns MD simulation. The right panel shows the structures extracted from four biased MD simulations using the harmonic position restraints around the sampled local minima in the FES (*md*1/*md*2/*md*3/*md*4; defined in Table 2). Compared to unbiased MD simulation, this last method allows to sample *md*3 and *md*4 regions, and to probe their contribution to Raman/ROA spectra. According to the FES profile of M13, we identified three different $\{\phi_1, \phi_2\}$ energy minima regions. Nevertheless, since the global minimum is quite narrow but long, we performed two biased MD simulations within this region, *md*1 and *md*2, differing mostly in $\phi_2$. Fig 4 (top right—MD) shows then calculated average Raman and ROA spectra that were obtained using the unbiased MD generated structures (Fig 3, MD—black crosses) and their comparison to experimental data. The simulated spectra already agree well with experiment, indicating a good performance of used MD force field. Most of the experimental spectral bands are correctly reproduced with comparable accuracy (S >0.9 for Raman and S >0.8 for ROA) as in our previous works with monosaccharides [10, 11]. Next, we used structures extracted from the biased MD simulations, which efficiently sample local minima of the FES profile. In Fig 4 (bottom), we show the Raman/ROA spectra for each of the conformer regions

**Table 2. $\phi_1/\phi_2$ restrain values of M13.** Restrain values [rad] of $\phi_1/\phi_2$ glycosidic angles used in biased MD simulations of M13 yielding *md*1/*md*2/*md*3/*md*4 conformers.

|  | *md*1 | *md*2 | *md*3 | *md*4 |
|---|---|---|---|---|
| M13 | -0.8/0.6 | -1.0/-0.6 | -0.6/-2.8 | 0.8/0.5 |

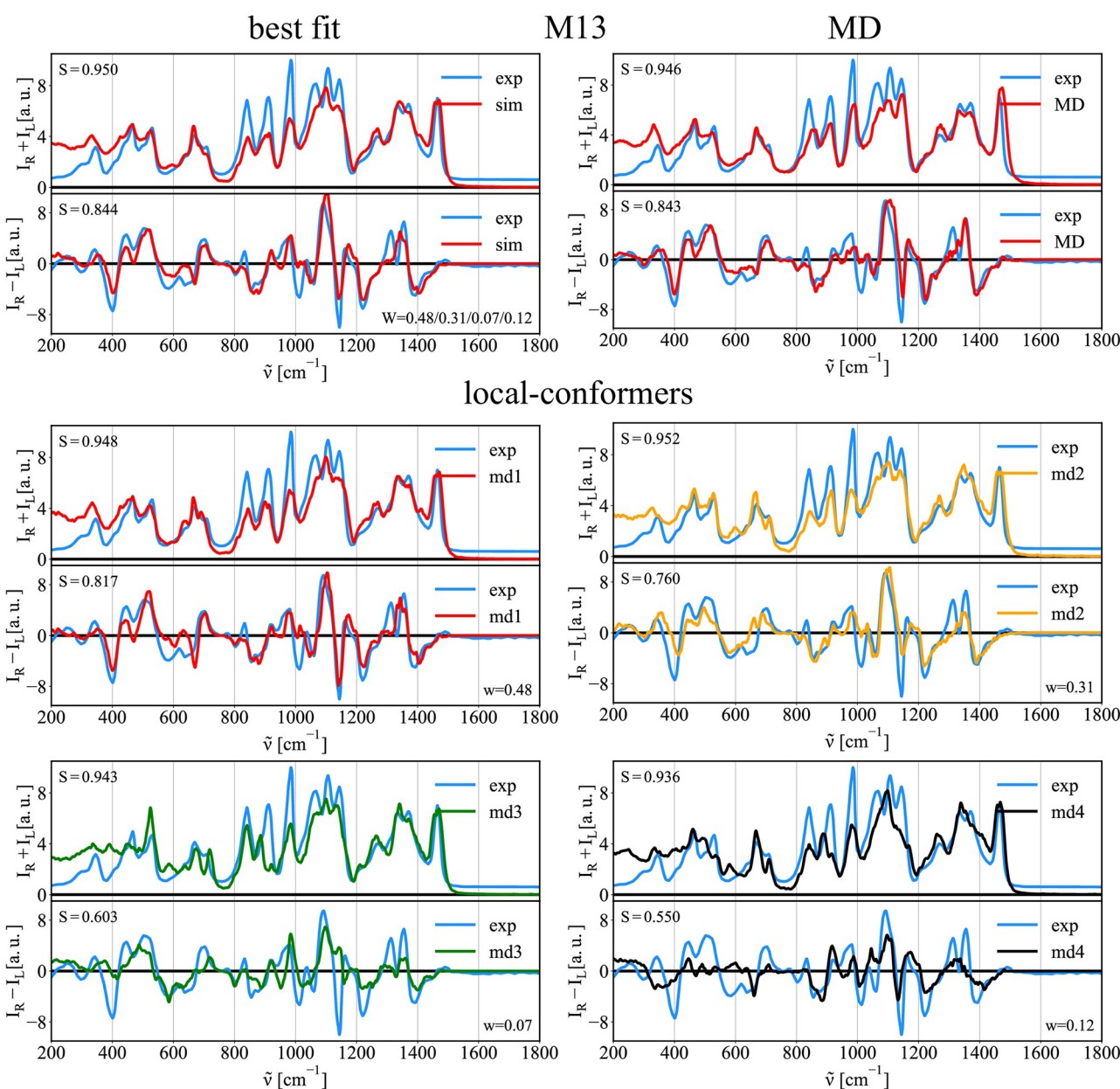

**Fig 4. Raman/ROA spectra of M13.** Comparison of experimental (exp.) and calculated spectra of M13 disaccharide. Top left—best fit: Best fit of *md*1/*md*2/*md*3/*md*4 Raman/ROA spectra to experimental data. Top right—MD: Simulated Raman/ROA spectra obtained using structures from unbiased MD simulation (MD). Bottom—local-conformers: Calculated ensemble averaged Raman/ROA spectra of M13 disaccharide prepared in 4 distinct conformations *md*1/*md*2/*md*3/*md*4 as described in Fig 3.

(*md*1/*md*2/*md*3/*md*4). Subsequently, we calculated the best fit spectra combining and weighting each conformers spectra to experimental data (Fig 4, best fit). Firstly, by inspecting the spectra of a selected conformer region (local-conformer), we see that the sensitivity of the Raman spectra to changes of the glycosidic angles is rather limited as the overlap integrals are in the range of $S_{Raman} = 0.936 - 0.952$ (0.948/0.952/0.943/0.936). On the other hand, the ROA spectra are much more affected $S_{ROA} = 0.550 - 0.817$ (0.817/0.760/0.603/0.550). Most significant differences between the conformers are concentrated in spectra below 900 cm$^{-1}$ and they

happen both in the Raman and ROA spectra. Interestingly, although the Raman spectra for frequencies >900 cm$^{-1}$ are almost identical, there are significant changes in ROA spectra. Finally, our fitting procedure (best fit) provides simulated spectra in better agreement ($S_{Raman}$ = 0.950 and $S_{ROA}$ = 0.844) with the experimental data than any conformer spectra. Only the *md*2 Raman spectrum alone agrees slightly better than best fit, but its ROA counterpart is far worse. This observation justifies the need of simultaneous fitting of the Raman and ROA spectra as this approach is more constrained and selective.

Although the MD simulation spectra and the best fit spectra provide almost identical accuracy (MD: $S_{Ram}$ = 0.951, $S_{ROA}$ = 0.838; best fit: $S_{Ram}$ = 0.948, $S_{ROA}$ = 0.844), they represent different conformer ratios. While the best fit data corresponds to *md*1/*md*2/*md*3/*md*4 = 0.48/ 0.31/0.07/0.12, the MD simulation gives 0.57/0.40/0.00/0.03. We observe an increase of *md*3 and *md*4 conformers, which MD misses. These differences can arise due to the equilibration issues (i.e., not long enough simulations) or because of inaccurate force field parameters. However, further analysis of obtained best fit ratios showed that there is a significant error in each conformer contribution estimation. Table 3 shows the best fit conformer ratios, together with error estimates of the fitting procedure described in the Methods section. The conformer ratios are determined with a 20–30% uncertainty, making effectively the Raman/ROA spectroscopies blind to contributions below this threshold. Nevertheless, by fitting the Raman/ROA spectra, we can distinguish all major conformers that have to be present in order to reproduce experimental data.

We also estimated the *md*1/*md*2/*md*3/*md*4 populations using the NMR $^3J_{CH}$ spin-spin coupling constants [41] and assessed the associated error. To calculate $^3J_{CH}$ from our simulations, we applied two approaches. In the first, we calculated the $^3J_{CH}$ coupling constants using the Karplus equation [41] for each frame in the bias MD simulations. In the second, we obtained the $^3J_{CH}$ coupling constants at the DFT level for the same set of frames, but after the QM optimization. Average $^3J_{CH}(sim)$ values were then calculated for each of the *md*1/*md*2/*md*3/*md*4 conformers, and then we performed the fitting of simulated values to experimental data [16], together with the error estimation (see Table F and G in S1 File for details of the fitting results). When using both approaches, we did not obtain a single best fit solution, but rather a range of values that fit the experimental data. This is because only two $^3J_{CH}$ coupling constants contain information about the glycosidic bond and are therefore fitted. The best fit population ranges are given in Table 3. Surprisingly, assuming 0.2 Hz deviation from the best fit solutions leads in both cases to large deviations making the NMR and Raman/ROA populations comparable, and burdened by comparable errors.

Considering the large uncertainty obtained when fitting the NMR and Raman/ROA experimental data, the MD populations of local-conformations probably reflect reality well. All methods that we used predict that the *md*1 conformer is the most abundant one (40–70%), followed with the *md*2 conformer (20–40%) with possible minor contributions of *md*3 and *md*4

**Table 3. M13 conformer populations.** Abundance of conformers of M13 disaccharide as obtained by MD, NMR, and the best fit to Raman/ROA (error bars in brackets).

| M13 conformer | MD | NMR (Karplus) | NMR (QM) | Best fit |
|:---:|:---:|:---:|:---:|:---:|
| *md*1 | 0.57 | 0.61–0.73[a] (0.25–0.89) | 0.43–0.55[a] (0.26–0.67) | 0.48 (0.18–0.89) |
| *md*2 | 0.40 | 0.00–0.23[a] (0.00–0.55) | 0.00–0.42[a] (0.00–0.56) | 0.31 (0.00–0.67) |
| *md*3 | 0.00 | 0.16–0.18[a] (0.04–0.29) | 0.00–0.28[a] (0.00–0.37) | 0.07 (0.00–0.35) |
| *md*4 | 0.03 | 0.00–0.09[a] (0.00–0.21) | 0.04–0.30[a] (0.00–0.41) | 0.12 (0.00–0.35) |

[a]no unique solution—values in this range fit experimental data

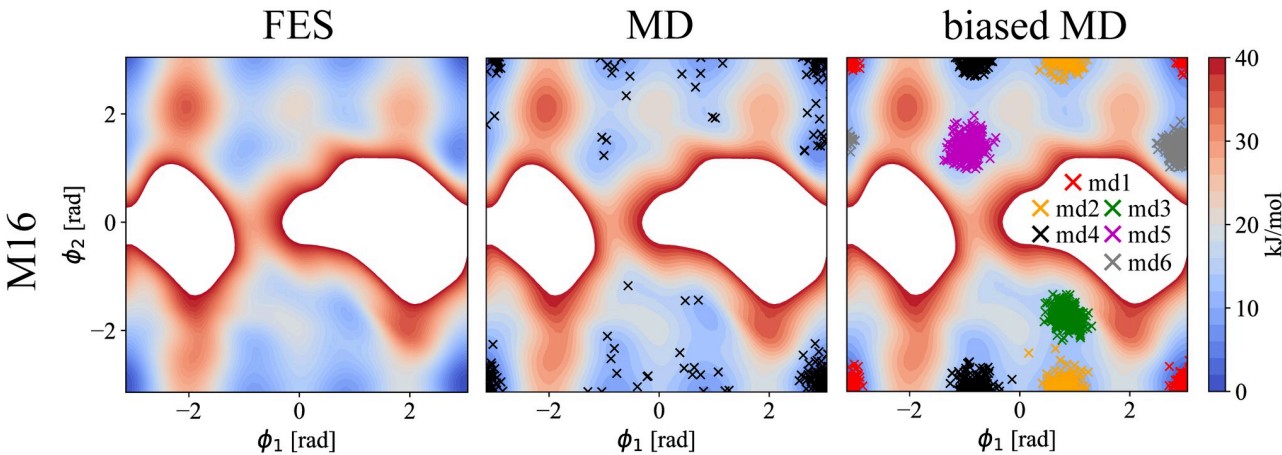

**Fig 5. M16 free energy surface and sampled structures.** Left: Calculated free energy surface (FES) of M16 disaccharide in terms of the $\{\phi_1, \phi_2\}$ dihedral angles. Middle: Calculated FES, together with 250 extracted structures from unbiased 500 ns MD simulations (MD). Right: Calculated FES, together with 250 extracted structures per each biased 200 ns MD simulation (biased MD; $md1/md2/md3/md4/md5/md6$; restrain $\{\phi_1, \phi_2\}$ values in Table 4). White regions represents area with the free energy >40 kJ/mol.

conformers. Evaluating contributions of conformers with such small weights (0–15%) is inconclusive as the uncertainty of abundances obtained by NMR/Raman/ROA are of the similar scale, and at least around 20%. Similar conclusions are drawn when studying trehalose and M13 disaccharides indicating a good performance of the Glycam force field for 1→1, 1→2, and 1→3 bonded disaccharides, at least in the mannose and glucose series. However, the additional degree of freedom in the 1→6 bonded M16 is more complicated, and it will be discussed in the next section.

**Conformation of methyl-1α-6α-mannobiose (M16).** As compared to the previously studied disaccharides, the free energy surface (FES) of M16 is unique. The linkage of the two mannose units consists of three bonds, and therefore, it requires three unique dihedral angles $\{\phi_1, \phi_2, \phi_3\}$ to describe its conformation. Fortunately, the $\phi_3$ angle population is rather confined around $\phi_3 \sim -0.84$ rad with a sharp profile with a full width at half maximum of 0.42 rad (See Fig E in S1 File). We therefore integrated this variable obtaining a 2D free energy profile ($\phi_1$ vs. $\phi_2$), see Fig 5. The obtained free energy profile has several accessible minima as compared to M12, M13, or trehalose disaccharides. Therefore, instead of sampling four regions as with M13/M12/trehalose, we sampled six local minima ($md1/md2/md3/md4/md5/md6$; defined in Table 4; depicted in Fig 5-right). For each of the minima region, we calculated corresponding Raman and ROA spectra. Fig 6 shows the Raman and ROA spectra of M16 calculated using structures from our MD simulation (MD), as well as by performing the best fit of local-conformer spectra to experimental data (best fit). As for M13, the Raman spectra of M16 present less variations compared to their ROA counterparts when comparing local-conformer spectra. Also, while several experimental bands do not manifest for some conformers (e.g., $\sim$ 600 cm$^{-1}$ or $\sim$ 850 cm$^{-1}$ bands), it is difficult to select particular bands as conformational probes. Again, major differences are in the lower frequency region below 1100 cm$^{-1}$

**Table 4. $\phi_1/\phi_2$ restrain values of M16.** Restrain values [rad] of $\phi_1/\phi_2$ glycosidic angles used in biased MD simulations of M16 yielding $md1/md2/md3/md4/md5/md6$ conformers.

|  | *md*1 | *md*2 | *md*3 | *md*4 | *md*5 | *md*6 |
|---|---|---|---|---|---|---|
| M16 | 3.14/3.14 | 0.8/-3.14 | 0.9/-1.7 | -0.8/-3.14 | -0.9/1.4 | 2.9/1.3 |

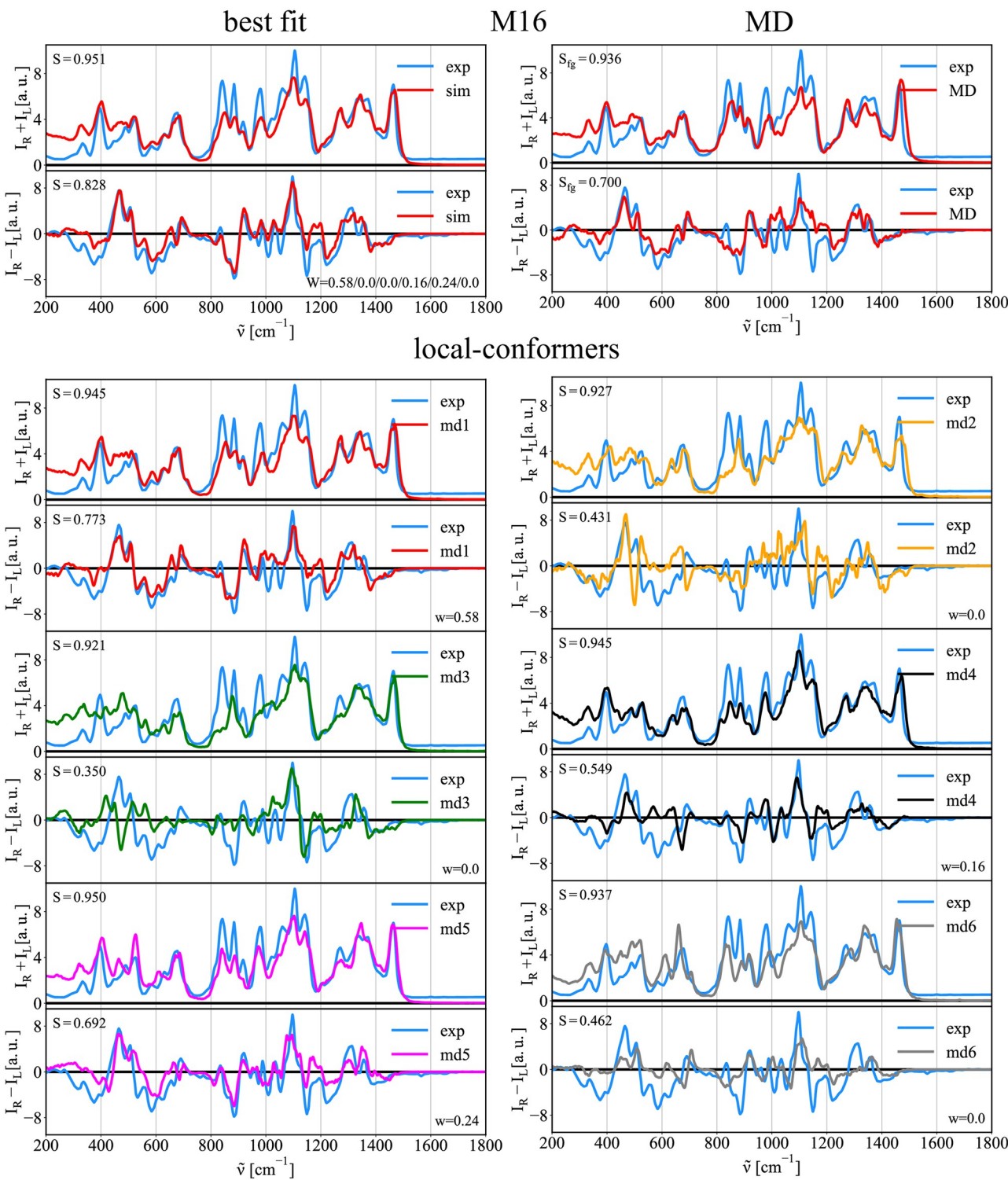

**Fig 6. Raman/ROA spectra of M16.** Comparison of experimental (exp.) and calculated spectra of M16 disaccharide. Top left—best fit: Best fit of *md*1/*md*2/*md*3/*md*4/*md*5/*md*6 Raman/ROA spectra to experimental data. Top right—MD: Simulated Raman/ROA spectra of disaccharide obtained using structures from unbiased MD simulation (MD). Bottom—local-conformers: Calculated ensemble averaged Raman and ROA spectra of disaccharide prepared in 6 distinct conformations *md*1/*md*2/*md*3/*md*4/*md*5/*md*6 as described in Fig 5.

**Table 5. M16 conformer populations.** Conformer abundances of M16 disaccharide obtained from MD and the best fit to Raman/ROA experimental data. Estimated error ranges for the latter in parenthesis.

| M16 conformer | MD | Best fit |
|:---:|:---:|:---:|
| $md1$ | 0.74 | 0.58 (0.36;0.78) |
| $md2$ | 0.08 | 0.00 (0.00;0.15) |
| $md3$ | 0.02 | 0.00 (0.00;0.13) |
| $md4$ | 0.07 | 0.17 (0.00;0.33) |
| $md5$ | 0.02 | 0.25 (0.00;0.48) |
| $md6$ | 0.07 | 0.00 (0.00;0.15) |

where oscillations of the whole sugar are probed. We found that the best fit solution outperforms the MD spectra ($S_{Ram}$ = 0.951 and $S_{ROA}$ = 0.828 for the best fit versus $S_{Ram}$ = 0.936 and $S_{ROA}$ = 0.700 for MD). This likely points out a force field inaccuracy in the description of the M16 conformational behavior. The conformer ratios of MD vs. the best fit significantly differs, contrary to disaccharides with a one-atom junction, i.e., M13, M12, and trehalose. The MD populations of selected regions, local-conformer, are $md1$/$md2$/$md3$/$md4$/$md5$/$md6$ = 0.74/0.08/0.02/0.07/0.02/0.07, while the best fit provides populations of 0.58/0.00/0.00/0.16/0.24/0.00. We find that the best fit solution identify as significantly populated three conformers, i.e., $md1$, $md4$, and $md5$, contrary to MD simulation, where a single conformer ($md1$) dominates. However, calculated population error ranges are significant (10–20%) with all results summarized in Table 5. Calculated error ranges are similar to those previously discussed for M13 or M12 pointing to a fundamental limitation of the Raman/ROA spectral analysis in terms of estimation of the conformer populations. Straight connection to NMR data is more difficult for M16 as compared to other studied disaccharides, e.g., for M13, since experimental data are available only for the $\phi_1$ dihedral angle [52]. Information about other two dihedral angles require for example $^3J_{CH}$ coupling constant across those glycosidic bonds requiring isotopically labeled compounds, which are harder to prepare. Since we do not have an access to such data, our NMR-based analysis of conformations is only limited. Nevertheless, some comparison of calculations with experiment is possible using $\phi_1$ only. Since $\phi_1$ is very similar for $md1$ and $md6$, $md2$ and $md3$, and $md4$ and $md5$ (see Fig 5), we can compare their sums. Analysis of available experimental $^3J_{HH}$ coupling constants using the Karplus equation can estimate the populations of $md1$ + $md6$ = 0.49, $md2$ + $md3$ = 0.06, and $md4$ + $md5$ = 0.45. Integrating our results show that MD provides $md1$ + $md6$ = 0.81, $md2$ + $md3$ = 0.10, and $md4$ + $md5$ = 0.09. The best fit to the Raman and ROA spectra results into $md1$ + $md6$ = 0.58, $md2$ + $md3$ = 0.00, and $md4$ + $md5$ = 0.42. The results are summarized in Table 6. Improving on the MD method, the best fit method to Raman/ROA provides essentially the same values as NMR experiments, even after the errors considerations.

**Table 6. M16 conformer populations.** Comparison of conformer populations of M16 disaccharide obtained using MD, NMR, and the best fit to experimental Raman/ROA data.

| M16 conformer | MD | NMR [52] | Best fit |
|:---:|:---:|:---:|:---:|
| $md1$+$md6$ | 0.81 | 0.49 | 0.58 |
| $md2$+$md3$ | 0.10 | 0.06 | 0.00 |
| $md4$+$md5$ | 0.09 | 0.45 | 0.42 |

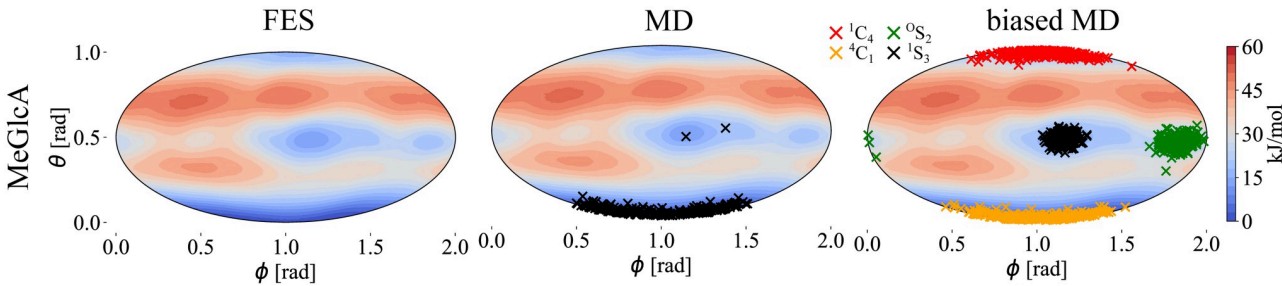

**Fig 7. MeGlcA free energy surface and sampled structures.** Left: Calculated FES of MeGlcA in $\phi/\theta$ puckering coordinates. Middle: Calculated FES, together with 250 extracted structures from unbiased 500 ns MD simulations (MD). Right: Calculated FES, together with 250 extracted structures per each biased 200 ns MD simulation(biased MD; $^1C_4/^4C_1/^OS_2/^1S_3$; restrain $\{\phi, \theta\}$ values in Table 7). All plots are shown as equal area Mollweide projection.

## Probing puckering conformation

The Raman and ROA spectra also allow probing of the puckering of saccharides, i.e., the ring accessible conformers. We will discuss the conformation of the pyranose ring further, but the approach is generally applicable also to saccharides with furanose rings. Fig 7 shows calculated free energy surface (FES) for methyl-β-D-glucuronic acid (MeGlcA) in the angular puckering coordinates, the MD sampled structures, and the structures gathered using the biased MD around the four selected local free energy minima($^1C_4/ ^4C_1/^OS_2/^1S_3$; defined in Table 7). MeGlcA free energy surface obtained using the Glycam force field reveals one deep global minimum around the south pole ($^4C_1$; $[\phi, \theta]$ of $[1, 0]$). Mostly all structures sampled from unbiased MD simulation falls on that region as well. Still, we probe other local minima ($^1C_4$, $^OS_2$, and $^1S_3$) as to whether they contribute to the final Raman/ROA spectra which could potentially points towards a force field inaccuracy. Fig 8 shows the Raman/ROA spectra of MeGlcA obtained for the unbiased MD, and using the best fit procedure. It also shows the spectra for each selected local-conformer each with different puckering coordinates, and their comparison with experiment. The unbiased MD spectra (both Raman and ROA) are practically identical to the best fit spectra (top right), which is composed purely from the $^4C_1$ spectra (bottom). All other conformers ($^1C_4/^OS_2/^1S_3$) do not contribute to the final spectra at all as summarized in Table 8. Again, we also tried to connect our MD/Raman/ROA results to available experimental NMR data [43]. To do this, we calculated ensemble average $^3J_{HH}$ spin-spin coupling constants for considered $^1C_4/^4C_1/^OS_2/^1S_3$ conformers at the QM level of theory (see Methods for details). Having obtained these, we performed the best fit to experimental data and then we estimated the error ranges of the fit (Table 8). Obtained populations are $^1C_4/^4C_1/^OS_2/^1S_3 = 0.01/0.99/0.00/0.00$, which is in a good agreement with both the MD and the best fit to Raman/ROA results. For the detailed values of the experimental and calculated $^3J$ spin-spin coupling constants see Table H in S1 File.

**Table 7. $\phi/\theta$ restrain values of MeGlcA.** Restrain values [rad] of $\phi/\theta$ puckering coordinates used in biased MD simulations of methyl-β-D-glucuronic acid yielding $^1C_4/ ^4C_1/^OS_2/^1S_3$ conformers. $^1C_4$ and $^4C_1$ conformer were restrained only in $\theta$ puckering variable.

|  | $^1C_4$ | $^4C_1$ | $^OS_2$ | $^1S_3$ |
|---|---|---|---|---|
| MeGlcA | all/3.14 | all/0 | 3.14/1.57 | 1.8/1.57 |

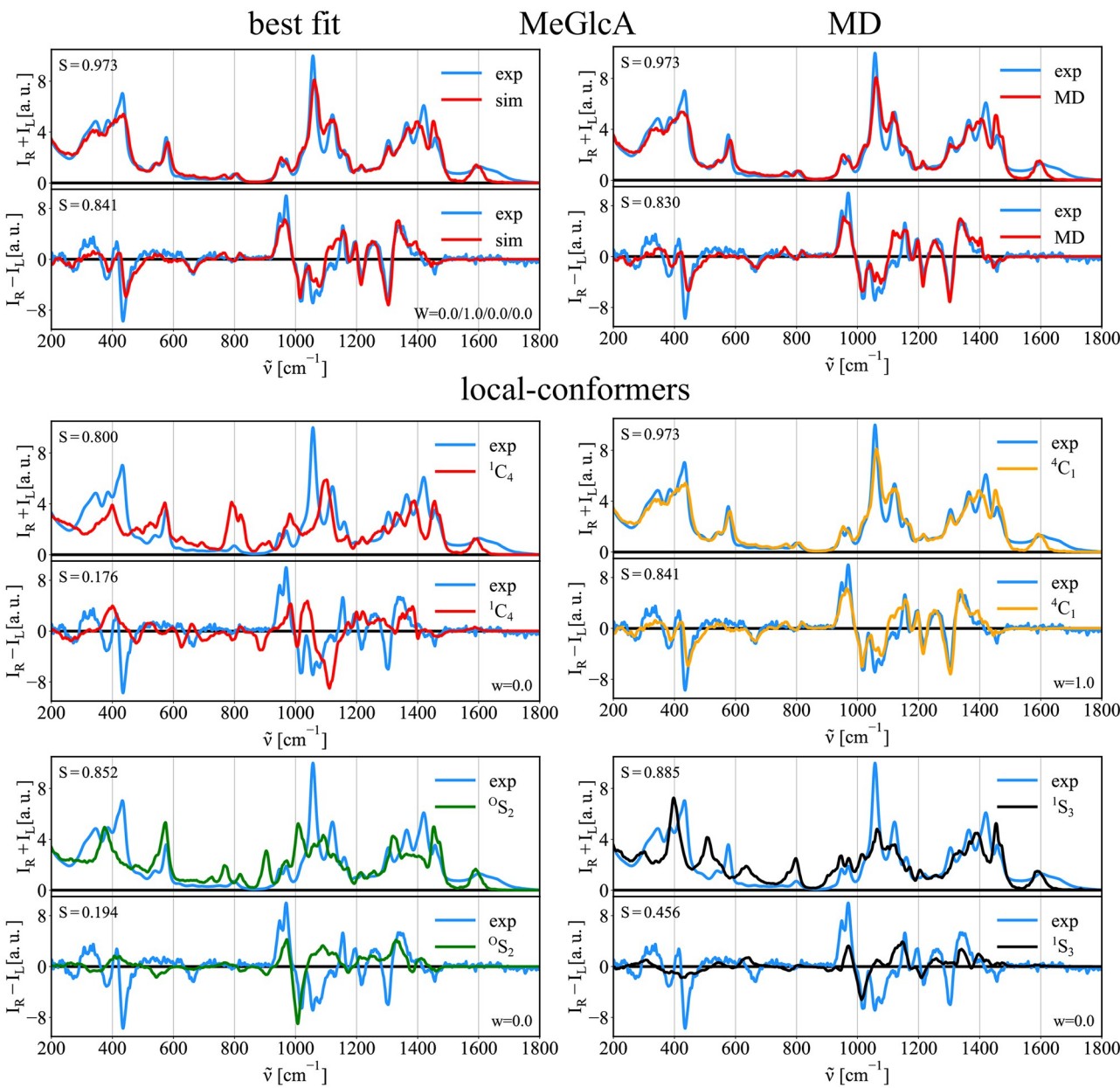

**Fig 8. Raman/ROA spectra of MeGlcA.** Comparison of experimental (exp.) and calculated spectra of MeGlcA. Top left—best fit: Best fit of $^1C_4/^4C_1/^OS_2/^1S_3$ Raman/ROA spectra to experimental data. Top right—MD: Simulated Raman/ROA spectra of the monosaccharide obtained using structures from unbiased MD simulation (MD). Bottom—local-conformers: Calculated ensemble averaged Raman and ROA spectra of the monosaccharide prepared in 4 distinct conformations $^1C_4/^4C_1/^OS_2/^1S_3$ as described in Fig 7.

## Raffinose trisaccharide

Study the structure of larger saccharides in solution is a major goal of structural biology. Since the used computational protocol was developed on monosaccharides [10], its performance in larger saccharides is not guaranteed. Therefore, we test the accuracy of the protocol here on a raffinose trisaccharide (See Fig 2) and discuss the raffinose structure based on the comparison of MD and experimental data. The MD calculations suggest that the molecule is found mainly

**Table 8. MeGlcA puckering conformer populations.** Abundance of puckering conformers of MeGlcA as obtained by MD, NMR, and the best fit to Raman/ROA experimental data. Estimated error ranges for the NMR and the best fit approaches are in parenthesis.

| MeGlcA conformer | MD | NMR[a] | Best fit |
|---|---|---|---|
| $^1C_4$ | 0.01 | 0.01 (0.00–0.03) | 0.00 (0.00–0.08) |
| $^4C_1$ | 0.98 | 0.99 (0.96–0.99) | 1.00 (0.91–1.00) |
| $^OS_2$ | 0.00 | 0.00 (0.00–0.03) | 0.00 (0.00–0.08) |
| $^1S_3$ | 0.01 | 0.00 (0.00–0.03) | 0.00 (0.00–0.08) |

[a]exp data fit [43]

in a few states (see Fig F-I in S1 File), where the most variable degrees of freedom are the $\phi_1$ and $\phi_2$ angles between both pyranoses (see Fig F in S1 File). Using structures obtained from the unbiased MD simulation, we calculated the Raman/ROA spectra given in Fig 9. A very good agreement with experiment was found considering the size and complexity of the system. The Raman spectrum is reproduced excellently with small overestimation of intensities at low frequencies around 250 cm$^{-1}$. These deviations, however, are also found for mannobioses, glucose, and glucuronic acid. The ROA spectrum is of slightly worse quality with the overlap integral of 0.733. Still, most features are unambiguously reproduced. Despite raffinose being a rather complex molecule, our results put into relevance the strength of our combined methodology to extract relevant structural features from sugars. We did not even need more advanced computational methods, e.g., biased MD, to obtain reasonable outputs for this molecule.

## The anomeric ratio

All reducing monosaccharides are in solution in equilibrium between their acyclic, and cyclic forms (e.g., pyranose/furanose). The reducing monosaccharides tested here (Glc, GlcNAc, and GlcA) are expected, according to NMR studies [53], to be found overwhelmingly in the cyclic form. This cyclic form of reducing saccharides in turn exist in an anomeric equilibrium between the α and β anomeric forms. While the contributions of the linear and furanose forms are negligible for our hexoses, the proportions of the α and β anomers are often comparable. Similarly to previous simulations, we can calculate pure spectra of the α and β anomers independently. We can then obtain the proportions of each anomer, i.e., anomeric equilibrium

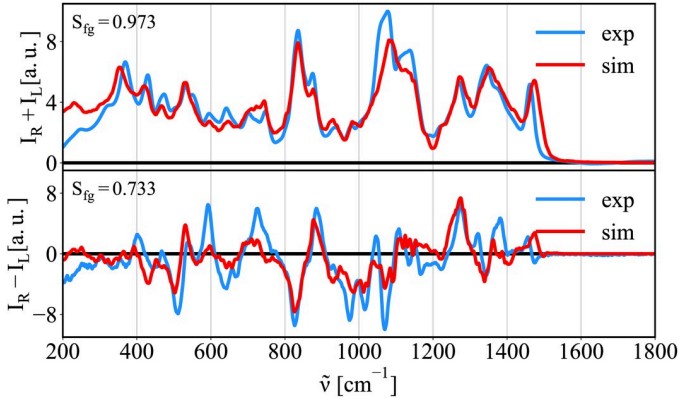

**Fig 9. Raman/ROA spectra of raffinose trisaccharide.** Comparison of calculated Raman and ROA spectra of raffinose trisaccharide with experiment.

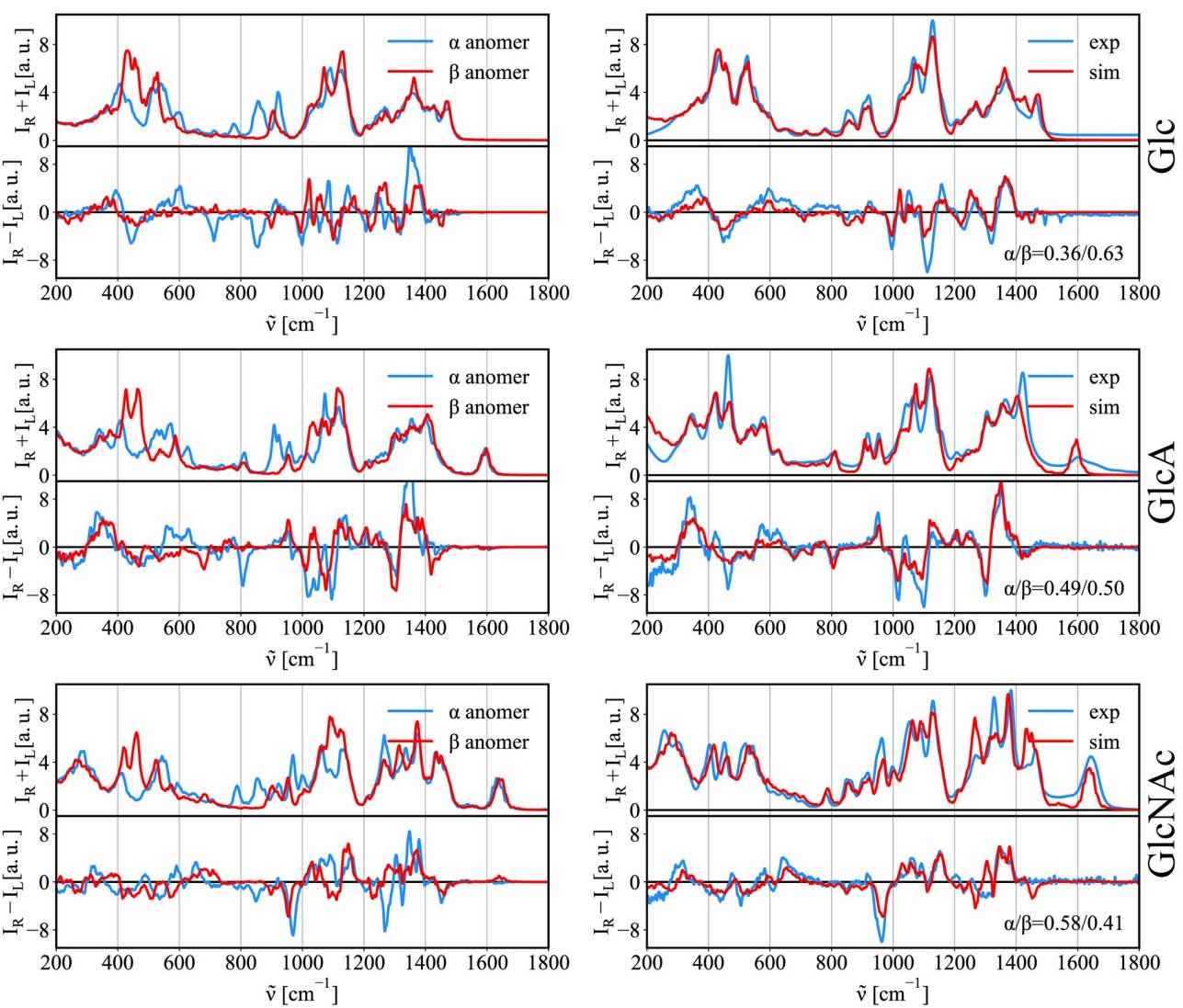

**Fig 10. Raman/ROA spectra of Glc, GlcA, and GlcNAc.** Left: Calculated Raman/ROA spectra for the α/β anomers (Glc, GlcA, and GlcNAc). Right: Best fit to experimental data.

value, by the linear fitting of calculated α/β anomer spectra to the experimental naturally mixed spectra. Fig 10 shows simulated Raman and ROA spectra of the α/β anomers (left) and the best fit to experimental data (right).

Table 9 summarizes calculated anomeric equilibrium ratio of Glc, GlcA, and GlcNAc. We find a reasonable agreement with experimental NMR data with exception of glucuronic acid,

**Table 9. Calculated anomeric ratios of Glc, GlcA, and GlcNAc.** Experimental and calculated anomeric ratios (β fraction) for investigated reducing sugars. Calculated data were obtained as the best fit of the Raman and ROA spectra to experimental data.

|     | Glc | GlcA | GlcNAc |
| --- | --- | --- | --- |
| Exp | 0.62–0.65 [53–55] | 0.60–0.62 [56, 57] | 0.35–0.42 [58] |
| Sim | 0.63 (0.49–0.76) | 0.50 (0.33–0.68) | 0.40 (0.24–0.56) |

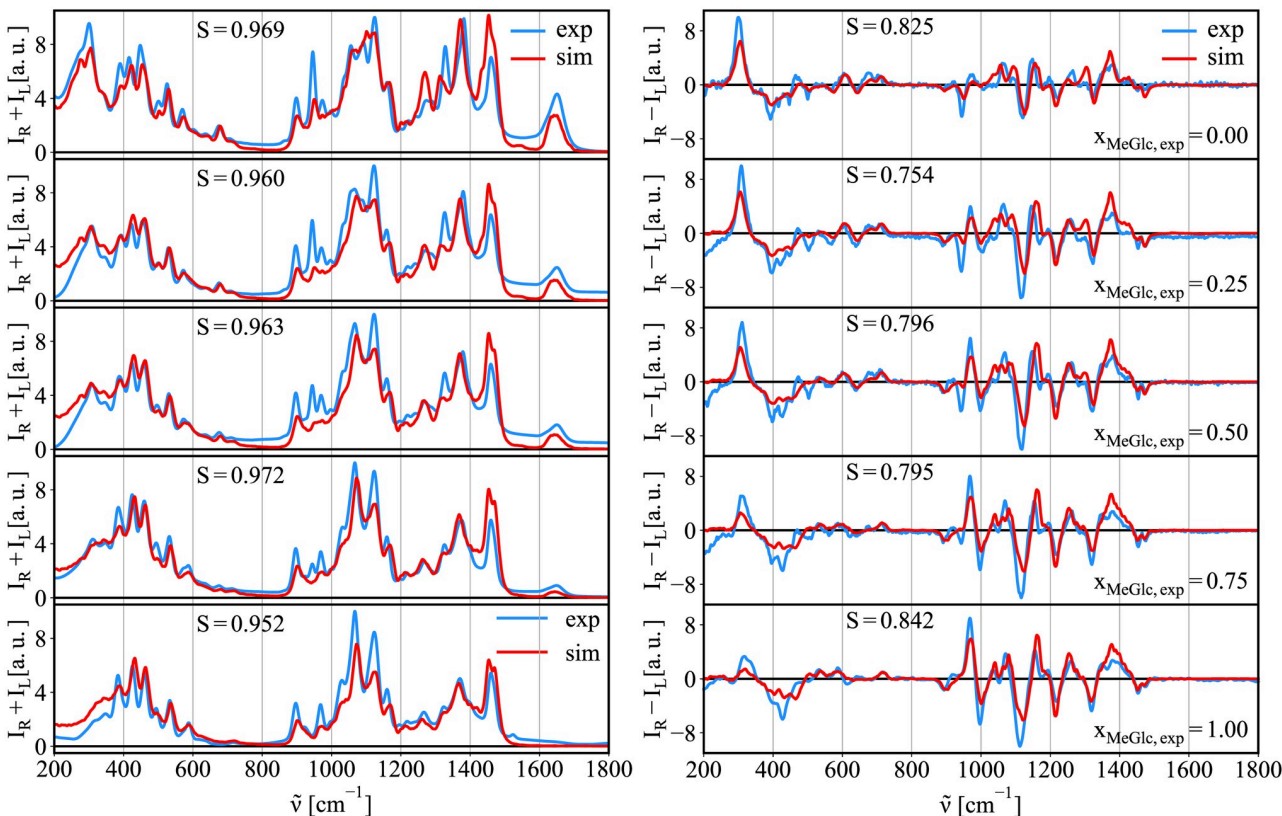

**Fig 11. Raman/ROA spectra of mixtures of MeGlc:MeGlcNAc.** In blue the experimental Raman/ROA spectra of MeGlc and MeGlcNAc, and their 3:1, 1:1, and 1:3 mixtures (MeGlc:MeGlcNAc). The best fit simulation spectra to the experiment using the spectra of simulated pure substances to fit them are shown in red (see Fig 12 and Table 10 for the results of the fit).

where we see an error of 10%. Similarly as in previous cases, we show the error estimates in brackets. We can conclude that predicted anomeric ratios based on the Raman and ROA spectra are much closer to the real experimental values than traditional predictions based on Boltzmann populations obtained for ab initio energies of each anomers that provide predictions with an error >15% [13].

Although we are able to find a reasonable agreement with experimental data, the error of the calculation is significant ($\sim$ 15%).

## Binary mixtures

We have discussed that it is possible to decompose experimental spectra to contributions of diferent chemical moieties and conformers, i.e., to calculate the anomeric equilibrium. We are not limited to the anomeric equilibrium, but any mixture can be studied in a similar manner using the same protocol. To exemplify this, we estimated the ratio of components in the experimentally prepared mixtures of methyl-β-glucose (MeGlc) and methyl-β-N-glucosamine (MeGlcNAc). Mixtures were prepared in 1:3, 1:1, and 3:1 ratios of MeGlc and MeGlcNAc. The pure substances were used as the references. Thus, the spectra of pure saccharides were calculated and used to decompose experimentally measured spectra ($x_{MeGlc}$ = 0.00, 0.25, 0.5, 0.75, 1.00). To probe the sensitivity of the methods, we have chosen chemical moieties with small

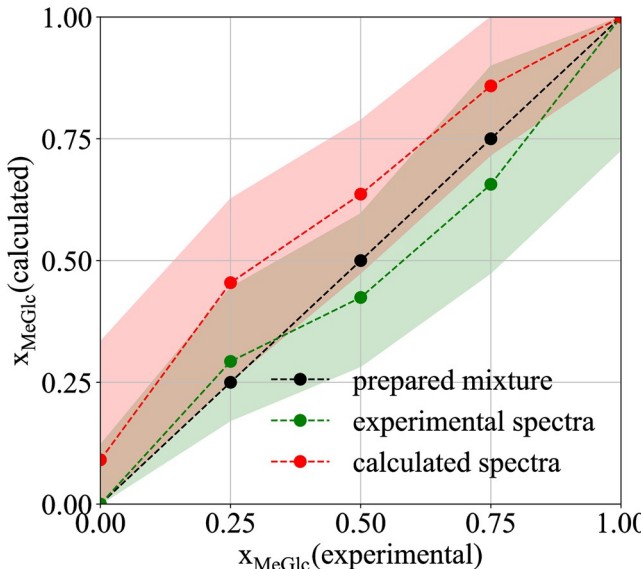

**Fig 12. Prediction of molar fractions of mixtures of MeGlc:MeGlcNAc.** Summary of calculated molar fractions and estimated errors of mixtures (MeGlc:MeGlcNAc) obtained by the best fitting corresponding experimental Raman/ROA spectra of known composition (black, $x_{MeGlc}$ = 0.00, 0.25, 0.5, 0.75, 1.00) using simulation(red, calculated spectra) or experimental(green, experimental spectra) spectra of pure substances to fit them.

structural differences. Therefore, we aim to capture relatively small differences in spectra (see simulated spectra of pure substances in section 5 in S1 File). Fig 11 shows spectra of the decomposition using simulation spectra of pure substances to fit the experimentally prepared mixtures. To compare, we also used experimental spectra of pure substances for the same fit. The results are summarized in Fig 12 and Table 10.

In all cases, despite the molecular similarities, the experimental molar fraction was reproduced when using the simulated spectra of pure substances for the fit, with just an slight overestimation of MeGlc ranging between ∼0–22% (error). The determined upper boundary composition error is larger than the error reported in ref. [13], where authors predicted the glucose/mannose ratio based on the ROA decomposition with an error <15%. This can be attributed to more significant differences in the structure and conformational behavior of glucose and mannose than what we see here for MeGlc and MeGlcNAc. The errors are also slightly larger than those computed for the anomeric equilibrium predictions discussed above. Still, when we evaluate the uncertainty of the best fit, the molar fractions can change up to 15%. Within such a error range, our calculations cover the real concentration, see Fig 12. Using experimental spectra of pure substance instead of simulated ones for the fit results in smaller errors, i.e., ∼ 0 − 9%. However, the evaluated uncertainties of the fit are ∼ 15%, which makes the simulation and experimental fits well comparable.

**Table 10. Calculated molar fractions of MeGlc:MeGlcNAc mixtures.** Summary of calculated molar fractions and estimated errors of prepared mixtures (MeGlc:MeGlcNAc; $x_{MeGlc}$ = 0.00, 0.25, 0.5, 0.75, 1.00) obtained by the best fitting corresponding experimental Raman/ROA spectra using simulation($x_{MeGlc}^{sim}$) or experimental ($x_{MeGlc}^{exp}$) spectra of pure substances to fit them.

| $x_{MeGlc}$ | 0.00 | 0.25 | 0.50 | 0.75 | 1.00 |
|---|---|---|---|---|---|
| $x_{MeGlc}^{sim}$ | 0.10 (0.00–0.33) | 0.47 (0.25–0.63) | 0.65 (0.48–0.78) | 0.87 (0.72–1.00) | 1.00 (0.90–1.00) |
| $x_{MeGlc}^{exp}$ | 0.00 (0.00–0.12) | 0.29 (0.17–0.44) | 0.42 (0.28–0.60) | 0.66 (0.47–0.90) | 1.00 (0.73–1.00) |

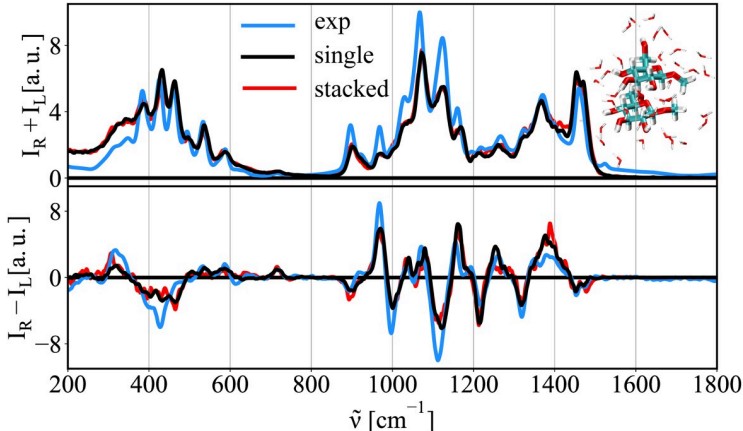

**Fig 13. Raman/ROA spectra of two MeGlc in close proximity.** Calculated Raman/ROA spectra of methyl-β-glucose at infinite dilution, i.e., single molecule (black), of two interacting methyl-β-glucose sugar moieties (red, representative snapshot in the inset). In blue the experimental spectra at 1 M concentration for comparison.

## Crowding effects

Crowded environments are common in living organisms, e.g., in their glycocalyx. This region is believed to be a relatively densely packed sugar region, where saccharides can potentially interact with each other. Here, we investigate whether such an interaction, i.e., the proximity of other sugar moiety and a contact with it, has a visible effect on the correspondent Raman/ROA spectra. Fig 13 shows a comparison of calculated Raman/ROA spectra of methyl-β-D-glucose in infinitely diluted solution (i.e., a single molecule, black), when in a close proximity of other sugar unit (stacked, red), and experimental data (blue) for comparison. The outcome is that both spectra are nearly identical in the studied frequency range. This is showing that the Raman/ROA spectroscopies cannot readily identify potential intermolecular interactions in the used spectral range unless there is a major structural change, e.g., as shown for MeGclA, which is easily visible. Fig K in S1 File shows that including empirical dispersion interactions in the DFT calculations has no effect on the outcome of the calculations.

## Discussion

Conformational behaviour of saccharides in solution is still not well understood, mainly due to the lack of readily available and suitable structural techniques. In the current paper, we suggest the use of Raman/ROA spectroscopies, together with computer simulations to determine structural properties of saccharides in solution. We show how the Raman and mostly ROA spectra when interpreted using appropriate computational methods resolve various structural features of sugars, such as the glycosidic linkage conformation, the ring puckering, the anomeric equilibrium, and we also reveal whether they provide any information on intermolecular interactions. Furthermore, they can be effectively used to quantitatively resolve more complex sugar mixtures.

To study the glycosidic linkages, we simulated the Raman and ROA spectra of four disaccharides (trehalose and three methyl mannobioses($1\rightarrow2$, $1\rightarrow3$, and $1\rightarrow6$ linked)) using ensemble generated either by unbiased MD simulations, or by advanced sampling techniques. Just briefly, simulations of M12 and M13 provided similar conformational profiles with the global minimum at $\{\phi_1, \phi_2\}$ of (-45, 35) or (-45, 22), respectively, which agrees with previous studies [45, 59, 60]. On the contrary, the simulation of M16 reveals that the global minimum is

the extended structure with $\{\phi_1, \phi_2\}$ of (180, 180). All but the 1→6 mannobiose provide a good agreement with experimental spectra when using unbiased MD ensemble. This indicates the reliability of the Glycam force field used to describe the 1→1, 1→2, and 1→3 glcyosidic bonds in MD simulations. On the other hand, the 1→6 linked mannobioses' spectra were reproduced with lower quality, suggesting inaccurate ensemble provided by the MD simulations. Using advanced sampling techniques, we were able to significantly improve on the quality of the simulation spectra, and therefore, to better identify the true ensemble. An important lesson learned is that the Raman/ROA spectra of disaccharides change significantly with simple variations in glycosidic dihedral angles. The changes occur mostly in the ROA spectra and below 1200 cm$^{-1}$. However, additional small changes can be detected in the whole considered spectral range. Moreover, we found that there are no specific bands in both Raman and ROA spectra associated to particular glycosidic bond orientations on the studied molecules. This lack of correspondence was also reported on mannobioses [16], suggesting that this a the common scenario. However, we show that the spectral changes produced by different glycosidic angles are significant enough to determine populations of conformers in a given disaccharide. Therefore, despite single bands cannot be assigned to particular conformations, coupling experimental spectra to computer simulations can yield preferred populations in solution. Our calculations also show that the error in determining the conformer populations using the Raman/ROA data is substantial, reaching up to 30%. Nevertheless, we also find a similar accuracy for predictions based on traditional NMR experiments that usually use $^3J_{CH}/^3J_{HH}$ coupling constants across glycosidic bonds. Moreover, to fully describe conformational preferences of 1→6 linked mannobiose, more coupling constant across the glycosidic link would be needed, e.g., $^3J_{CH}/^3J_{CC}$ or $^2J_{CC}$ [61, 62], and not only the easily available $^3J_{HH}$. However, this is only achievable with isotopically $^{13}C$ labeled compounds, which are usually hard to synthesize. Our approach does not require any labelling, and using the Raman/ROA spectroscopies we are able to decompose the spectra into local-conformers (average of configurations around a conformation) even without the labeled compound. This could for example help in computational studies relying in conformational preferences on *C*-disaccharides, where the two saccharide units are joined with a methylene bridge, where little structural information is still known [63]. Overall, our data praises the Glycam force field that accurately describes the rotation around 1→1, 1→2, and 1→3 glycosidic bonds, while finding that the 1→6 linkage requires refinement.

We also tackle a basic structural feature of saccharides, their puckering, using methyl-β-D-glucuronic acid. We prepared models of the monosaccharide in various puckering conformations and subsequently calculated the Raman/ROA spectra for each puckering conformation. The variations in the Raman and ROA spectra produced by change of the puckering conformation were significant enough to assign unequivocally the $^4C_1$ puckering conformation as the dominant conformation of methyl-β-D-glucuronic acid, which is in agreement with NMR observations.

We also show that the `hybrid` calculation protocol, originally developed on monosaccharides [10], is also applicable to larger saccharides. These larger saccharides are the focus of many state of the art structural works focus on Raman/ROA experiments [64, 65]. With the use of MD and advanced sampling methods, we explored and described the phase space of one of these sugars, a raffinose trisaccharide. This a a challenging saccharide composed of three different monosaccharides and two likages. Still, our method can accurately reproduce experimental spectra, showing the applicability of the simulation protocol for trisaccharides at a reasonable computational cost. Still the overall accuracy is dependent on the simulation, and future improvements of the computational protocol are desirable.

As reducing sugars exist in solution as a balanced mixture of different forms, we addressed the sensitivity of the approach to discriminate such individual forms and to estimate their

occurrence. Therefore, we explored the fundamental anomeric equilibrium for glucose and two glucose derivatives. Both anomers of glucose, glucuronic acid, and N-acetyl glucosamine were prepared and corresponding Raman/ROA spectra were calculated. By the best fitting to experimental spectra, we were able to successfully predict the value of the α/β anomeric equilibrium constant with an error of $\sim 10\%$. The results match with NMR-based results, which is a more commonly used method for estimating the anomeric ratio. Further analysis showed that the best fit to the Raman/ROA data is associated with 10–15% uncertainty error, which is slightly smaller than the error found when determining populations of disaccharide conformers. This smaller error might be attributed to the fact that puckering changes the whole geometry of a molecule, leading to very distinct Raman/ROA spectra.

It was already shown that the spectral decomposition approach can be applied to estimate the composition of saccharide mixture consisting of two different diastereomers [13]. This has also been applied successfully to other sterols and complex rings mixtures in chlororoform [66, 67]. Here we go even further to test the sensitivity limit of the approach, and estimate the ratio of very similar sugars such as a monosaccharide and its amine. We decompose experimental spectra of mixtures of methyl-β-glucose (MeGlc) and methyl-β-N-glucosamine (MeGlcNAc). Note that MeGlcNAc differs from MeGlc only by substitution of the hydroxyl at carbon C2, and the achiral amide group should not generate an intrinsic ROA signal. We prepared samples of pure sugars, as well as their mixtures ($x_{MeGlc} = 0.25/0.50/0.75$). Then we experimentally recorded their Raman/ROA spectra and subsequently decomposed the spectra finding the ratio of molecular moieties in the target solution using either simulated or experimental spectra of the pure compounds. When using computed spectra, we were able to predict the compositions of mixtures within $\sim 0–22\%$ errors. These error naturally resembles the error for the anomeric equilibrium values, as both procedures are in essence identical. When using experimental spectra of pure substances for the decomposition, we obtain better fits, i.e. $\sim 0 - 9\%$ errors. However, when we evaluated uncertainties of the bets fit values for both cases, they are equally footed. Lastly, this decomposition method is potentially applicable to other moieties, not just saccharides.

Saccharides in biology are often found in crowded environments. Therefore, we probe the sensibility of the Raman/ROA spectra to study intermolecular interactions usual in such environments. For this we calculate the spectra of a sugar moiety in the close proximity to another. When comparing a model system of a single methyl-β-glucose (MeGlc) with two MeGlc in a close proximity (average COM-COM distance 0.46 nm), we see that interaction has no effect on the Raman/ROA spectra in the studied 200–1800 $cm^{-1}$ spectral range. Therefore, any direct interaction using Raman/ROA cannot be easily seen, unless the interaction/packing results in a change of the geometry/structure of given saccharides upon the interaction. The result is in agreement with previous works in the literature, where it was concluded that the interaction regions is likely too small and that there could be some interaction specific bands present below 200 $cm^{-1}$ [68, 69].

Overall, in our work we demonstrate the power of coupling of the Raman/ROA spectroscopies with computer simulations. This allows to extract numerous atomistic details that would be otherwise hardly obtainable. We also quantified the expected reliability of our structural predictions, which despite having substantial associated error up to 41%, this errors are comparable standard techniques based on NMR spin-spin coupling constants.

## Conclusion

Simulation of Raman/ROA spectra of saccharides is becoming well established field. However, its applications in structure determination, in particular for large molecules, is yet largely

unexplored. In this paper, we show that combining ROA/Raman spectra with advanced sampling MD techniques and QM calculations, one can get vast amount of structural information about saccharides in solution. Taking advantage of the sensitivity of Raman/ROA to structural changes one can detect even small changes in rotation around the glycosidic bonds. This also applies for the puckering changes of the sugar rings. We show that the used simulation protocol can also applicable to a larger moiety, e.g., the raffinose trisaccharide, at a reasonable computational cost. We can also apply the methods to mixtures. We, for example, managed to extract α/β anomeric equilibrium values of monosaccharides. But the method can be extended to any sort of mixture, even for those structurally similar. We showed that Raman/ROA spectra in the $200 - 1600$ cm$^{-1}$ range are insensitive to direct contacts unless they produce structural changes. Lastly, the method used here allows to estimate its inherent error when determining populations, which in this study can be as high as 41%, and comparable with other simulation and experimental methods actively used. All investigated cases show that coupling computer simulations with experimental Raman/ROA spectroscopies is a powerful structural determination tool.

## Supporting information

**S1 File. Supplementary information file.** The file includes a thorough description of the simulated systems and used methods. Moreover, it includes additional and supporting data for the main paper.
(PDF)

## Acknowledgments

We acknowledge the European Regional Development Fund OP RDE (project ChemBioDrug no. CZ.02.1.01/0.0/0.0/16_019/0000729) for computational resources. We thank Prof. Pavel Jungwirth and Prof. Petr Bouř for fruitful discussions.

## Author Contributions

**Conceptualization:** Vladimír Palivec, Hector Martinez-Seara.

**Data curation:** Vladimír Palivec.

**Formal analysis:** Vladimír Palivec.

**Funding acquisition:** Hector Martinez-Seara.

**Investigation:** Vladimír Palivec, Jakub Kaminský, Hector Martinez-Seara.

**Methodology:** Vladimír Palivec, Christian Johannessen, Jakub Kaminský, Hector Martinez-Seara.

**Project administration:** Hector Martinez-Seara.

**Resources:** Hector Martinez-Seara.

**Software:** Vladimír Palivec.

**Supervision:** Hector Martinez-Seara.

**Visualization:** Vladimír Palivec.

**Writing – original draft:** Vladimír Palivec, Hector Martinez-Seara.

**Writing – review & editing:** Christian Johannessen, Jakub Kaminský.

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
