## [Decision Letter · Decision Letter 0]

15 Dec 2021

Dear Dr. Martinez-Seara,

Thank you very much for submitting your manuscript "Use of Raman and Raman optical activity to extract atomistic details of saccharides in aqueous solution" for consideration at PLOS Computational Biology. As with all papers reviewed by the journal, your manuscript was reviewed by members of the editorial board and by several independent reviewers. The reviewers appreciated the attention to an important topic. Based on the reviews, we are likely to accept this manuscript for publication, providing that you modify the manuscript according to the review recommendations.

Sincerely,

David van der Spoel

Associate Editor

PLOS Computational Biology

Nir Ben-Tal

Deputy Editor

PLOS Computational Biology

[LINK]

Reviewer's Responses to Questions

**Comments to the Authors:**

Reviewer #1: In their manuscript, the authors present their methodology of improved calculations of Raman, and Raman optical activity spectra of carbohydrates in aqueous solution and analysis of their ring puckering conformations, anomeric ratios, or glycosidic bonds, extended for a bigger systems, e.g. disaccharides and trisaccharides. There are limited tools to study carbohydrate structure in a native aqueous solutions. ROA/Raman tend to be very useful in this field, showing great potential, however analysis based on theoretical calculations are challenging. Their work is very laborious and impressive. Combination of Raman/ROA experiments with MD and QM based calculations, together with modern approach of partial optimization of snapshots, scaling functions or best fit procedure, gave great results of the similarity index of the experimental and simulated spectra, and extensive information of the carbohydrate structure. Although this work is suitable for publication in the PLOS Computational Biology, several questions and concerns arises after reading, that are listed below.

1) Please describe precisely, point by point what is the novelty of present manuscript compared to former paper from Palivec et al. PCCP.,2020, 22, 1983 (10.1039/c9cp05682c). My first impression was that it is to some point extend of the former paper to di- and trisaccharides. Please clarify in the manuscript, what is new here and what is based on the previous paper.

2) To understand properly all methods used here, I read the authors' previous work (Palivec et al. PCCP.,2020, 22, 1983), together with the Reply to Reviewers that was added as the SI. After careful reading, I still have concerns about scaling of calculated intensities. In the previous paper to which the reader is referred, you mentioned “The average magnitudes of ROA and Raman intensities are experimentally related (I(ROA)~I(Raman)x10^-4), however, we optimize both spectral intensities separately. As a result, the ratio is not necessarily preserved during the optimization”, Is it the case also in this paper? If yes it should be mentioned somewhere, and discussed, especially because all simulated and experimental ROA/Raman spectra are presented here in some puzzling arbitrary units, where ROA and Raman intensities are comparable, although one can expect rather ~10^-4 ratio of ROA/Raman intensities. In my opinion it will be puzzling for future readers. I have a problem with it because you are losing information about CID ratios. As we all know, CID ratios are inherently associated with ROA, and their calculation and comparison with experiment can improve the reliability of structural conclusions (Polavarapu, CHIRALITY 26:539–552 (2014)). At least I would suggest that the original ROA/Raman ratios of both experimental and simulated spectra should be preserved during all the scaling, and minimization of the cost function. I’m curious how would it change the final results?

3) About bibliography. I would suggest to enrich the bibliography with the latest papers on ROA of trisaccharides, or polysaccharides e.g. 10.1039/C9CP00472F, or 10.1016/j.saa.2018.08.017, where in the latter, raffinose ROA spectrum is discussed.

Reviewer #2: Please see the report file.

**Have the authors made all data and (if applicable) computational code underlying the findings in their manuscript fully available?**

Reviewer #1: None

Reviewer #2: None

PLOS authors have the option to publish the peer review history of their article (what does this mean?). If published, this will include your full peer review and any attached files.

Reviewer #1: No

Reviewer #2: No

Figure Files:

Data Requirements:

Reproducibility:

References:

---

## [Editor Report · Decision Letter 1]

3 Jan 2022

Dear Dr. Martinez-Seara,

We are pleased to inform you that your manuscript 'Use of Raman and Raman optical activity to extract atomistic details of saccharides in aqueous solution' has been provisionally accepted for publication in PLOS Computational Biology.

Best regards,

David van der Spoel

Associate Editor

PLOS Computational Biology

Nir Ben-Tal

Deputy Editor

PLOS Computational Biology

---

## [Editor Report · Acceptance letter]

17 Jan 2022

PCOMPBIOL-D-21-02118R1 

Use of Raman and Raman optical activity to extract atomistic details of saccharides in aqueous solution

Dear Dr Martinez-Seara,

I am pleased to inform you that your manuscript has been formally accepted for publication in PLOS Computational Biology. Your manuscript is now with our production department and you will be notified of the publication date in due course.

With kind regards,

Livia Horvath
